# Forecast of Winter Precipitation Type Based on Machine Learning Method

**DOI:** 10.3390/e25010138

**Published:** 2023-01-10

**Authors:** Zhang Lang, Qiuzi Han Wen, Bo Yu, Li Sang, Yao Wang

**Affiliations:** 1Chongqing Research Institute of Big Data, Peking University, Chongqing 400000, China; 2China Huaneng Clean Energy Research Institute, Lab. Building A, Huaneng Innovation Base, Future Science Park, Beiqijia Town, Changping District, Beijing 102209, China; 3Beijing Weather Forecast Center, Beijing 100097, China; 4Academy for Advanced Interdisciplinary Studies, Peking University, Beijing 100871, China

**Keywords:** winter precipitation-type prediction, machine learning, model output statistics

## Abstract

A winter precipitation-type prediction is a challenging problem due to the complexity in the physical mechanisms and computability in numerical modeling. In this study, we introduce a new method of precipitation-type prediction based on the machine learning approach LightGBM. The precipitation-type records of the in situ observations collected from 32 national weather stations in northern China during 1997–2018 are used as the labels. The features are selected from the conventional meteorological data of the corresponding hourly reanalysis data ERA5. The evaluation results of the model performance reflect that randomly sampled validation data will lead to an illusion of a better model performance. Extreme climate background conditions will reduce the prediction accuracy of the predictive model. A feature importance analysis illustrates that the features of the surrounding area with a –12 h offset time have a higher impact on the ground precipitation types. The exploration of the predictability of our model reveals the feasibility of using the analysis data to predict future precipitation types. We use the ECMWF precipitation-type (ECPT) forecast products as the benchmark to compare with our machine learning precipitation-type (MLPT) predictions. The overall accuracy (ACC) and Heidke skill score (HSS) of the MLPT are 0.83 and 0.69, respectively, which are considerably higher than the 0.78 and 0.59 of the ECPT. For stations at elevations below 800 m, the overall performance of the MLPT is even better.

## 1. Introduction

There are three phases of winter precipitation: liquid, solid, and liquid–solid mixed. Different types of precipitation have different effects on the substance circulation and energy exchanges. For example, liquid precipitation changes the energy partition of sensible heat and latent heat by affecting the water content of surface soil, but solid precipitation such as snowfall alters the energy budget by affecting the surface albedo before it melts [1,2]. Solid precipitation with the same 24-hour accumulated precipitation amount of drizzle may be classified as heavy snow which may cause meteorological disasters. Rain or sleet falls that on the ground below 0 °C will freeze and may lead to hazardous road surface conditions [3]. Thus, precise in-time knowledge on precipitation types is crucial for local transportation management, road maintenance, and aviation ground deicing operations in winter.

Several studies on the formation mechanism of precipitation types have shown that the most critical factor affecting the phases of precipitation is the temperature profile [4,5,6,7,8,9]. The melting layer with a temperature above 0 °C and the subfreezing layer in the atmosphere directly determine the phase state of the precipitation. Ice particles that pass through the melting layer may partially melt and turn into sleet or wet snow, or completely melt and turn into rain on the ground. If there is a subfreezing layer near the surface due to cold advection, these completely or partially melted precipitation particles may freeze again and become ice pallets on the ground. Glickman and Zenk [10] gave specific definitions for winter precipitation types.

Although the precipitation type is mainly influenced by the atmospheric temperature profile, it is also affected by the particle falling speed, particle size, surface latent heat, atmospheric radiation, and other factors [11]. For example, because small-scale particles are more likely to melt completely in the melting layer than large particles, the precipitation types formed by these two kinds of particles are completely different [9,12]. Thus, there continue to be many considerable uncertainties within precipitation-type forecasting. The accurate prediction of precipitation types is still one of the most challenging problems in winter precipitation forecasting [13,14].

The traditional methods of precipitation-type forecasting are mainly based on numerical weather prediction models (NWP). In addition to improving the physical parameterization schemes of the NWP, some statistical techniques were developed to optimize the prediction [9]. For example, Bocchieri [15] developed the Model Output Statistics (MOS) technique for forecasting the conditional probability of precipitation types. They used the pressure-layer thickness, boundary-layer potential temperature, specific temperature, dew-point temperature, and wet-bulb temperature as the predictors to build a regression function with station precipitation-type records. Keeter and Cline [16] analyzed the relationships between a 1000–700, 850–700, and 1000–850 hPa thickness and precipitation types by using a stepwise linear regression and provided an MOS with additional objective precipitation-type guidance. Bourgouin [17] developed the so-called area method to diagnose the surface precipitation types. The area in this method is defined as the area between the temperature profile of a melting (refreezing) layer and the 0 °C isotherm on an aerological diagram and is considered to be proportional to the resident time of precipitation particles.

Previous studies assumed that the relationship between precipitation types and meteorological elements is linear. However, many physical processes of the atmosphere are nonlinear and require nonlinear solutions. Some machine learning methods were applied to solve the nonlinear problems of precipitation-type forecasting by optimizing the results of the NWP. For example, some studies used decision trees and logistic regression to identify the types of precipitation based on meteorological variables [18,19,20,21,22]. Moon and Kim [2] used multinomial logistic regression to improve the forecast accuracy based on the outputs of the NWP. Most machine learning forecast methods construct predicting features using near-surface meteorological elements, which cannot well represent the vertical profile of atmospheric heat and water vapor that play key roles in the physical process of the phase change in the precipitation; in addition, an algorithm includes the regional historical climate information as the prediction feature, which may restrict its performance due to the generalization issue of the algorithm over space.

In this paper, we propose an innovative forecast method for precipitation types using machine learning. Besides commonly used near-surface variables such as the 2 m temperature and dew-point temperature, the prediction features adopted in this method also include the vertical profiles of the temperature, relative humidity, wind speed, and vertical velocities from 500 to 1000 hpa.

Yang et al. [23] compared the differences between three machine learning algorithms (XGBoost, SVM, and DNN) for a precipitation-type diagnosis using analysis and forecast data. They concluded that the diagnosis results of all three methods based on the analysis data, which are more accurate than forecast data due to the assimilation of observations, have a higher accuracy. In addition, there were some successful machine learning applications of weather forecasting [24,25,26,27,28]. It is reasonable to speculate that if we use truth data to predict the precipitation phase directly, the prediction may have a higher accuracy than the NWP. In this study, we also discussed the prediction ability of the machine learning approach based on the in situ observations of the conventional meteorological data from 32 Chinese national weather stations.

This paper is organized as follows. Section 2 introduces the datasets used in our study. Section 3 illustrates the machine learning approach and prediction evaluation method. Section 4 expands on model performances. In Section 5, we provide the discussion of the results. Finally, Section 6 presents the conclusions.

## 2. Materials

### 2.1. Observations of Precipitation Types

The in situ observations of precipitation types used in this paper are from 32 Chinese national weather stations (NWS) in Beijing and surrounding areas (hereafter referred to as BJA). The weather phenomena are manually recorded every three hours. They are coded into 100 categories, and approximately 30 are related to precipitation events. We collected and recoded the precipitation events during winter months (January, February, March, November, and December) of 1997 to 2018 (data from only three stations in 1997) into three categories, i.e., rain, sleet, and snow. The frequency of each recoded precipitation type is shown in Figure 1 with different symbol sizes. The altitude (represented by color in Figure 1) of the 32 stations ranges from 30 to 1400 m, covering both plains and mountains. Figure 1 also shows that rain rarely occurs in mountain areas during winter in BJA, sleet seldom appears in plain areas, and snow is the main precipitation type in this area during winter.

### 2.2. Meteorological Data

ERA5 is the fifth-generation ECMWF atmospheric reanalysis of the global climate covering the period from January 1950 to present. It integrates vast amounts of historical observations into global estimates using advanced model and data assimilation system and thus can be used as pseudo observations in meteorological studies [29]. Because long-term observed pressure-level data are not available, we use the pressure-level data of ERA5 as an approximation to the truth. For data consistency, we also choose suface-level data of ERA5 data to substitute for surface observations. Hourly data of ECMWF forecasting productions with a spatial resolution of 0.125${}^{\circ }\times 0.125^{\circ }$ are also used in our study as a benchmark measure of model performance.

## 3. Methods

### 3.1. Feature Preparation

The goal of supervised learning is to build a prediction model for the distribution of class labels in terms of predictor features [30]. The model is defined as
(1)y=F(x;w)
where *F* is the model, y is labels, x is predictor features, and w is model parameters trained by the machine learning algorithm. Commonly used predictive factors of precipitation-type predictions are pressure-level thickness, temperature of a specific layer, dew-point temperature, height of the 0 °C isotherm, wet-bulb temperature, etc. [31]. These predictors are not the results derived from NWP directly but are intermediate variables computed using conventional data (temperature, wind, pressure, and humidity). Some formulas for intermediate variables calculation are semiempirical and would bring uncertainties for precipitation-type diagnosing and forecasting. Hence, we consider training the prediction model by conventional data directly. Large-scale atmospheric circulation processes are the primary physical mechanism of the generation of winter precipitation in northern China. The advection and local changes in meteorological elements are crucial for forecasting these weather processes. We considered using data from the closest grid to the NWS and grids in adjacent regions because the advective changes in atmospheric states were difficult to be obtained from local information. Due to computing resource limitations, we only used nine grids in the adjacent area of the NWS, which were approximately in the 12.5 × 12.5 km horizontal resolution. The predictability time was restricted to less than 24 h due to the area limits, based on categories of atmospheric motions classified by scales. In this work, we construct a feature space with a total dimension of 9714, including 13 h × 16 pressure levels (ranging from 500 to 1000 hPa) × 3 × 3 grids of temperature (t), specific humidity (r), meridional wind component (v), zonal wind component (u), and vertical wind speed (w). In addition to these vertical elements, three surface meteorological variables: 2 m temperature (t2m), 2 m dew-point temperature (d2m), and surface pressure (sp) with dimensions of 13 h × 3 × 3 grids also used. The center point of the 3 × 3 grids is the closest grid to the NWS. Finally, three basic geographical parameters of the weather stations, including latitude, longitude, and altitude, are also constructed in the feature space.

A schematic diagram of the feature space is shown in Figure 2. We selected the data of Δh hours before the observation time to be the features for examining the predictability of the machine learning approach used in this paper.

### 3.2. Light Gradient Boosting Machine (LightGBM)

LightGBM is a gradient boosting framework that uses tree-based learning algorithms; thus, its hypothesis space is the tree model. It is designed to be distributed and efficient with the following advantages: faster training speed and higher efficiency; lower memory usage; better accuracy; support of parallel, distributed, and GPU learning; and capable of handling large-scale data [32]. It uses two novel techniques: Gradient-based One-Side Sampling (GOSS) and Exclusive Feature Bundling (EFB) which fulfills the limitations of the histogram-based algorithm that is primarily used in all GBDT (Gradient Boosting Decision Tree) frameworks. The two techniques of GOSS and EFB form the characteristics of the LightGBM algorithm. They comprise together to make the model work efficiently and provide it a cutting edge over other GBDT frameworks.

GOSS keeps those features with large gradients and only randomly drops those features with small gradients to retain the accuracy of information gain estimation. This procedure can lead to a more accurate gain estimation than uniformly random sampling, with the same target sampling rate, especially when the value of information gain has a large range. The variance gain at feature *j* can be calculated by
(2)$$\tilde{V_{j}}\left(d\right)=\frac{1}{N}\left(\frac{{\left(\sum_{x_{i}\in A_{l}}g_{i}+\frac{1-a}{b}\sum_{x_{i}\in B_{l}}g_{i}\right)}^{2}}{N_{l}^{j}\left(d\right)}+\frac{{\left(\sum_{x_{i}\in A_{r}}g_{i}+\frac{1-a}{b}\sum_{x_{i}\in B_{r}}g_{i}\right)}^{2}}{N_{r}^{j}\left(d\right)}\right)$$
where $A_{l}=\left\{x_{i}\in A;x_{ij}\leq d\right\}$, $A_{r}=\left\{x_{i}\in A;x_{ij}>d\right\}$, $B_{l}=\left\{x_{i}\in B;x_{ij}\leq d\right\}$, $B_{r}=\{x_{i}\in B;$$x_{ij}>d\}$, and *a* is the sampling ratio of large gradient data, *b* is sampling ratio of small gradient data, *d* is a point of the *j*th feature at which the decision tree model splits the data into the left and right child nodes, *x* represents the training data, $x_{i}$ is the *i*th instance and is a vector with s dimensions, s is the number of features, $x_{ij}$ is the *i*th instance of the *j*th feature, *g* denotes the negative gradients of the loss function, $N=\sum I[x_{i}\in A]$ is the size of training instances, $N_{l}^{j}=\sum I[x_{i}\in A:x_{ij}\leq d]$, $N_{r}^{j}=\sum I[x_{i}\in A:x_{ij}>d]$, A=a×100% instances is a subset with the larger gradients, *B* is a subset with data randomly sampled from the remaining set of *A* consisting of *b* samples with smaller gradients, the subscripts *l* and *r* indicate the left and right leaf nodes.

The Exclusive Feature Bundling (EFB) technique bundles the exclusive features into a single feature by designing a careful feature-scanning algorithm. EFB improves the speed for model training without sacrificing accuracy due to reducing the number of features.

Because LightGBM speeds up the training process of the conventional Gradient Boosting Decision Tree by up to over 20 times without sacrificing accuracy, we choose this algorithm to predict the winter precipitation types in this study.

### 3.3. Forecast Verification

The main forecasting skill of winter precipitation type is evaluated in terms of accuracy (ACC). Table 1 shows a contingency table for three categories of rain, sleet, and snow. ACC is defined as
(3)$$ACC=PC=\frac{a+b+c}{a+b+\cdots +h+i}$$

The meaning of each letter of this formula is shown in Table 1. However, when class imbalance is not negligible in the dataset, as is the case in our study, ACC is a biased estimate of model performance which may provide misleading results on classes with fewer samples. Thus, we employ another metric, the Heidke skill score (HSS) [33], to take the problem of class imbalance into account. The HSS, also known as kappa score in statistics, is “a skill score for categorical forecasts where the proportion correct (PC) measure is scaled with the reference value from correct forecasts due to chance” [34]. By definition, HSS is calculated using the following formula:(4)$$HSS=\frac{PC-\frac{(a+d+e)\times \left(a+f+h\right)+(f+b+g)\times \left(d+b+i\right)+(h+i+c)\times \left(e+g+c\right)}{{\left(a+b+\cdots +h+i\right)}^{2}}}{1-\frac{(a+d+e)\times \left(a+f+h\right)+(f+b+g)\times \left(d+b+i\right)+(h+i+c)\times \left(e+g+c\right)}{{\left(a+b+\cdots +h+i\right)}^{2}}}$$

### 3.4. Experiment Setup

We used the ECMWF atmospheric reanalysis data ERA5 and the NWS data from November 2016 to December 2018 as the test dataset. For model training, we designed two types of data schemes. One is to randomly split the remaining dataset into two parts, i.e., the training dataset and the validation dataset, with a ratio of 80%:20%. We randomly resampled five times to produce experimental dataset combinations (set1–set5). Considering that the actual forecasting operation always trains the predictive model by historical data and inspects the model by observations in a future period, the other scheme is designed to split the remaining dataset by periods. For example, the data from January 1998 to March 2000 are set as the validation dataset, and the data range from November 2000 to March 2016 is set as the training dataset. Detailed information on all 14 dataset combinations is shown in Table 2. We obtained nine different combinations (set6–set14) by sampling data with nine periods as the validation datasets. In addition, the Δh displayed in Figure 2 is set to be 0 to –12 h for precipitation-type diagnosis and exploring predictability. We obtained 14 prediction models and corresponding verifications based on these dataset combinations. Precipitation types in short-range forecasting products of ECMWF (ECMWF P-type) are used as a benchmark to evaluate the model performance of the test dataset. The period of ECMWF P-type data we used is the same as the test dataset.

## 4. Results

### Model Performance

The model performance of the 14 experiments with Δh=0 is shown in Table 3, depicting the evaluation results of the ACC and HSS based on the validation and test datasets. Three precipitation-type distributions of the validation datasets are also presented in Table 3. For the random sampling cases (Set1–Set5), both the ACC and HSS indicate the consistency of the models and have a good performance with an ACC > 0.9 and HSS > 0.8. However, the evaluation results obtained a lower ACC and HSS when the trained model was applied to predict the test data. The winter precipitation in northern China mostly lasts for a few hours. Therefore, randomly sampled validation and training data may originate from the same precipitation processes. It makes an illusion that the model performs better on the validation data and leads to the discrepancy of the model performance for the validation and test datasets.

For the time-split sets (Set6–Set14), the evaluation results based on the validation datasets range from 0.79 to 0.88 for the ACC and 0.48 to 0.73 for the HSS, but the ACC and HSS of the test datasets maintain a relatively steady performance level. Because of the steady performance on the test datasets, the trained model is considered to be stable, but the samples in the validation datasets may not be evenly distributed. The HSS indicates a more realistic accuracy of the evaluation from which the impact of random forecasts is eliminated. For classification problems, if the sample size of one category is too small, then the HSS will be low. We checked the validation samples of Set7–Set14, and only Set8, Set10, and Set14 had relatively balanced proportions of each category. Especially for Set11, only 5% of the validation samples are rain and obtain the lowest HSS.

Set6 (the bold type in Table 3) has the lowest evaluation scores for the validation data, and Set8 obtains the highest. The temperature profile statistical characteristics of the three precipitation types for Set6 to Set14 are shown in Figure 3. The vertical temperature structure for the three categories of precipitation type below 700 hPa of Set6 is distinctly different from the other sets. We checked the El Nino indices over the past decades to verify this speculation and then found that a very strong El Nino event occurred in 2015–2016 (set6). Although 1997–1998 is also a strong El Nino year, it does not have such high indices as 2015–2016. If precipitation events under extreme climate conditions are not involved in the training dataset, the trained model cannot perform well on the test data with similar cases. The above reason leads to the worst model performance for Set6. The strong La Nina of Set8 (the bold type in Table 3) in 2010–2012 is very similar to what occurred in 1998–1999 (Set14) and 2007–2008 (Set10). Hence, Set8 has the best model performance. We chose Set6 (a very strong El Nino year), Set7 (a normal year), Set8 (a strong La Nina year), and Set10 (a mixture of an El Nino and La Nina year) to analyze the Kernel Density Estimations (KDEs) of three relatively more important elements (the 850 hPa specific humidity, Figure 4a,d,g,j; 850hPa temperature, Figure 4b,e,h,k; and temperature at 2 m, Figure 4c,f,i,l). The shapes of the KDE for Set6 and Set8 display opposite patterns. In the very strong El Nino year (Set6), the air of 850hPa seems wetter and warmer for snow events and drier and colder for rain events. The KDE curves of the 2 m temperature in Figure 4c illustrate a similar pattern as displayed in Figure 4a,b. It is reasonable to conclude that the abnormal data shapes of the meteorological elements due to the anomalous circulation pattern in the extreme climate background limit the prediction ability of the MLPT.

To explore the predictability of the machine learning approach used in this paper, we set Δh to be from –1 to –12 h and then trained a predictive model based on a random sampling dataset. The test dataset is used to evaluate the forecast products of this predictive model. Figure 5 shows the evaluation results of the precipitation-type predictions and reveals the feasibility of using the analysis data directly to predict future precipitation types.

To analyze the terrain impact on our model prediction and retain enough sleet samples, we split the data of 14 sets into below 800m (69% of all stations) and above 800 m (31% of all stations) by the station altitude. Table 4 displays the ACC and HSS of the training models for stations at elevations below 800 m. The models based on the lower-elevation stations data perform significantly better than those based on the data for all stations except for Set6 and the validation datasets of Set1–Set5. This phenomenon demonstrates that the influence of topographic uncertainty on our model predictions is not as crucial as that of the sampling mode and extreme climate background. The performance of our prediction model is more stable for flat areas. The same predictability exploration mentioned in the previous paragraph based on the data from all stations is applied to assess the predictability of a model trained by the data from stations below 800 m. The assessment result is also shown in Figure 5. Unlike the characteristics of the results for all stations, the ACC and HSS become visibly lower as the forecast horizon becomes longer.

## 5. Discussion

### 5.1. Feature Importance

The feature importance reflects how useful the input features are at predicting a target variable and can provide insight into the dataset and the model. The feature space constructed in this study includes four categories of dimensions, i.e., the meteorological elements, vertical structure, temporal structure, and spatial structure. Figure 6 shows the average feature importance of 14 models along the temporal and spatial (vertical and horizontal) dimensions. The t2m and d2m are of high importance to the model. This reflects that the t2m and d2m of 3 h before the observation time contribute more to the model for distinguishing the precipitation types. The importance of the vertical velocity component is approximately uniform in all dimensions and is higher than that of the temperature and horizontal wind profile. As the thickness tendency can be evaluated by the vertical velocity [35], the basis of using the vertical velocity to classify precipitation types is essentially similar to using pressure thickness which is a classic traditional method of precipitation-type forecasting [4,15,16]. The temperature profile at the adjacent grid of the NWSs with a –12 h offset from the precipitation occurrence time is more important than that at other positions to the model. The winter precipitation in northern China is caused mostly by large-scale circulation. Therefore, the temperature structure of the surrounding area with a –12 h offset time may have a higher impact on the ground precipitation types. Specific humidity on upper layers with larger time offsets is more important than on lower layers with fewer time offsets.

### 5.2. Comparative Analysis

The comparison of the forecast precipitation types based on the MLPT with the ECPT is shown in Table 5. The MLPT shows a stronger predicting power on rain and sleet but slightly weaker on snow than the ECPT. The accuracy of each precipitation type for the MLPT and ECPT is shown in Figure 7a. Although the accuracy of the sleet prediction increases 10% compared to the ECPT, the ACC is too low so the forecast of sleet still has a small predictive significance. The overall ACC and HSS of the MLPT are 0.83 and 0.69, respectively, which are considerably higher than the 0.78 and 0.59 of the ECPT. For stations with elevations below 800 m, the overall ACC and HSS of the MLPT and ECPT are (0.86, 0.75) and (0.78, 0.62), respectively. The evaluation results of the MLPT at lower-elevation stations shown in Table 6 are better than those at all stations. However, the accuracy and HSS of the ECPT do not significantly improve. Thus, we consider that the advancement of the model performance at low-elevation stations is due to a reduction in the terrain impact but sample distribution variation. Figure 7 shows the ACCs of the three precipitation types for all stations and stations with elevations below 800 m. We can see in Figure 7 that removing the high-elevation stations data improves the predicting power of our model on snow. The false data of the pressure levels below surface pressure at high-elevation stations are the primary source of the topographic uncertainty of our method.

The logistic regression (LR), Supporting Vector Machine (SVM), Naive Bayes (NB), and Multi-layer Perceptron (MLP) are popular machine learning techniques. The LR and SVM were used to diagnose precipitation types in other studies [2,23]. We compared the model performance of the LightGBM with these four different schemes for predicting the precipitation type based on the training data of Set1 and the same test dataset. The results of the other schemes are shown in Table 7. The results show that the precipitation type based on the SVM is the closest forecast skills comparing to that of the LightGBM. Nevertheless, the LightGBM has a distinctive advantage of computing efficiency (1.48 s of LightGBM and 254.79 s of SVM for 10,000 forecasts).

## 6. Conclusions

We introduced a new approach for predicting the winter precipitation type based on machine learning techniques and examined the predictive power of this method. Previous methods of precipitation-type prediction used intermediate variables as the candidate predictors or features, such as the pressure-level thickness, dew-point temperature, isotherm height, wet-bulb temperature, etc. The method used in this paper chooses the vertical profile of routine meteorological variables between 500 and 1000 hpa as the features and can not only diagnose but also predict the precipitation type in 0–12 h.

Groups of datasets are designed for model training and a model performance evaluation. Based on these 14 groups of datasets, we have the following findings:If the validation data are random samples from the whole data, there would be an illusion of a better model performance because the validation and training data may originate from the same precipitation processes.The predictive model does not perform well in predicting the precipitation types under extreme climate conditions.The false data of the pressure levels below the surface pressure at high-elevation stations lead to the prediction error.The results of the model predictability exploration reveal the feasibility of using the analysis data to predict future precipitation types.

The overall ACC and HSS of our proposed model are 0.83 and 0.69, respectively. They are considerably higher than the 0.78 and 0.59 of the ECMWF forecasting products. Even though the model performance under an abnormal climate background is not so good (ACC = 0.79 and HSS = 0.58), the averaged prediction accuracy is still comparable to the forecasting skills of the forecast products of the ECMWF. After removing the high-elevation stations data, the overall ACC and HSS of the predictive model are 0.86 and 0.75, respectively. There is an increase of approximately 8 and 13%, respectively, in the ACC and HSS compared to the ECPT.

## Figures and Tables

**Figure 1 entropy-25-00138-f001:**
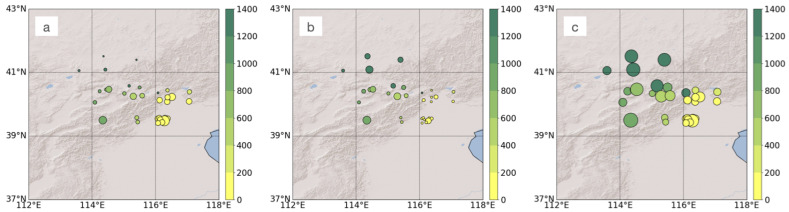
Distribution of the 32 Chinese national weather stations, with the diameter of the circle representing the frequency of rain (**a**), sleet (**b**), and snow (**c**), and the color representing altitude of the station.

**Figure 2 entropy-25-00138-f002:**
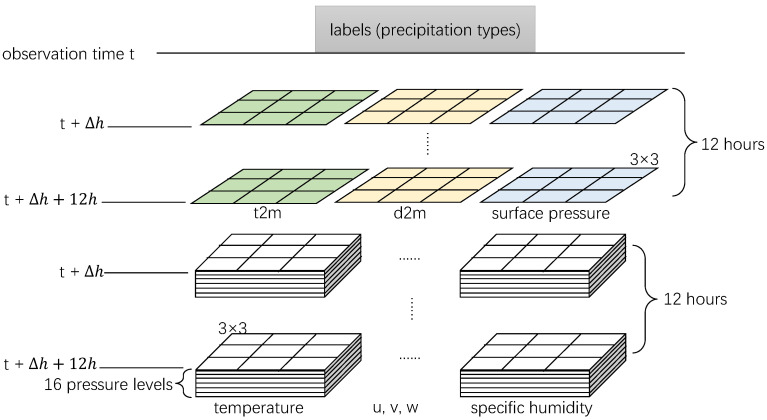
Schematic diagram of the feature space (without geographic parameters). Abbreviations correspond to those mentioned in Section 3.1, and Δh is the predictability time preset for the model.

**Figure 3 entropy-25-00138-f003:**
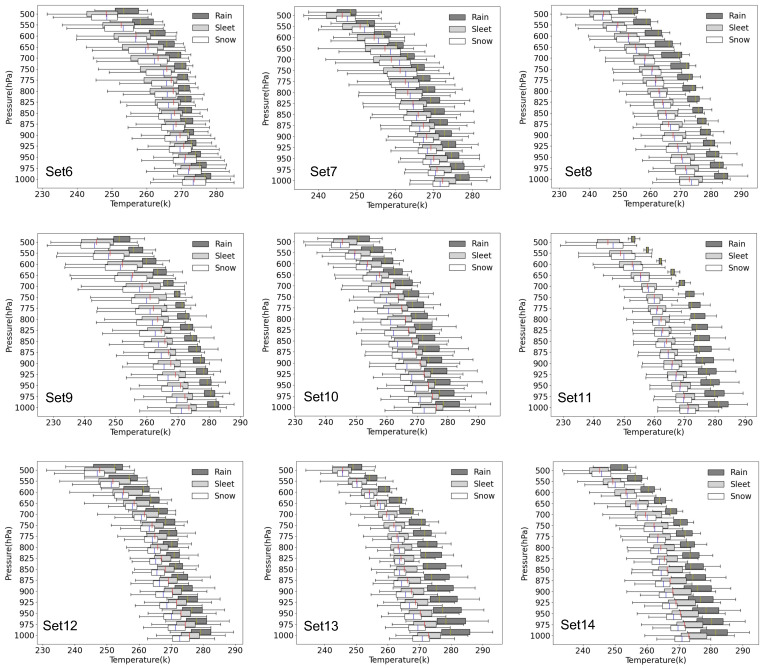
Distribution of air temperature for rain, sleet, and snow on pressure levels.

**Figure 4 entropy-25-00138-f004:**
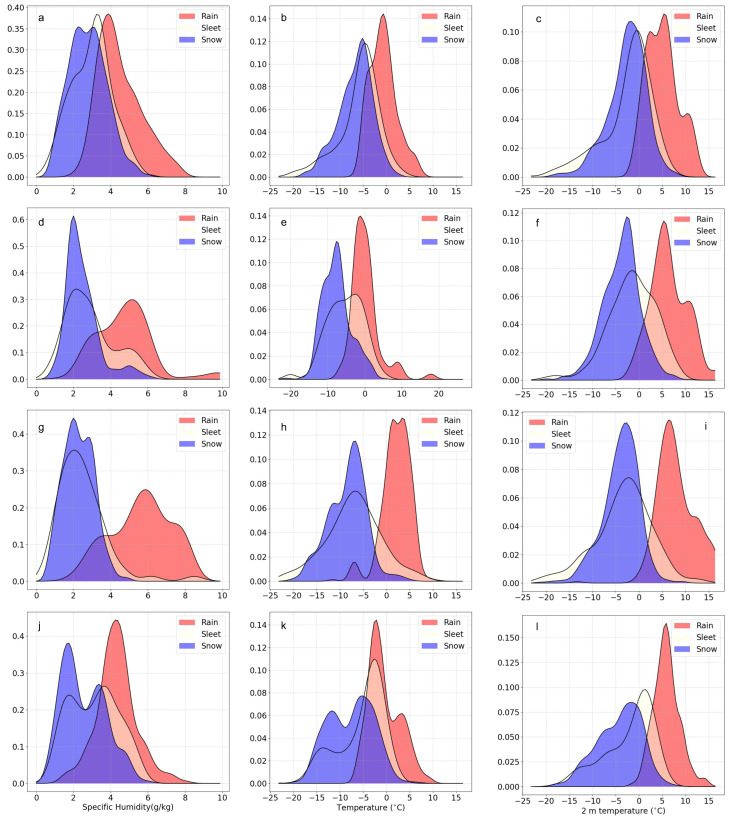
Kernel Density Estimation for rain, sleet, and snow according to specific humidity ((**a**,**d**,**g**,**j**) for set6–set8 and set 10), temperature at 850 hPa ((**b**,**e**,**h**,**k**) for set6–set8 and set 10), and temperature at 2 m ((**c**,**f**,**i**,**l**) for set6–set8 and set 10).

**Figure 5 entropy-25-00138-f005:**
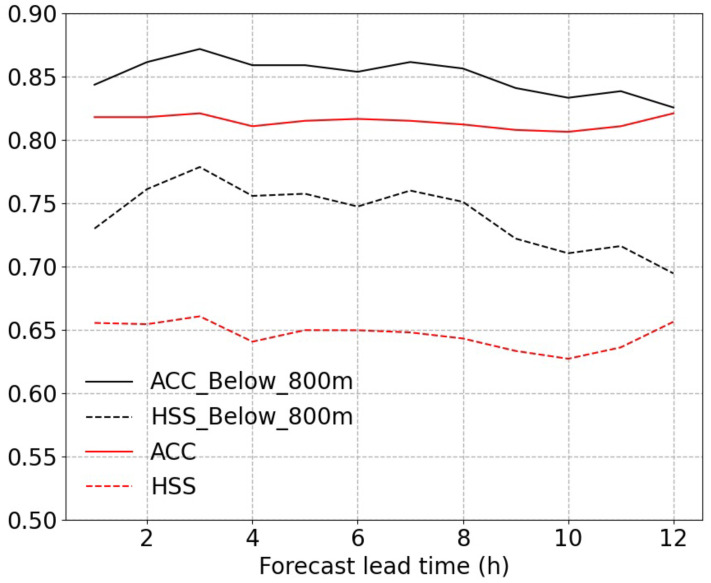
ACC and HSS of precipitation-type predictions at all stations and at stations below 800 m for different lead times.

**Figure 6 entropy-25-00138-f006:**
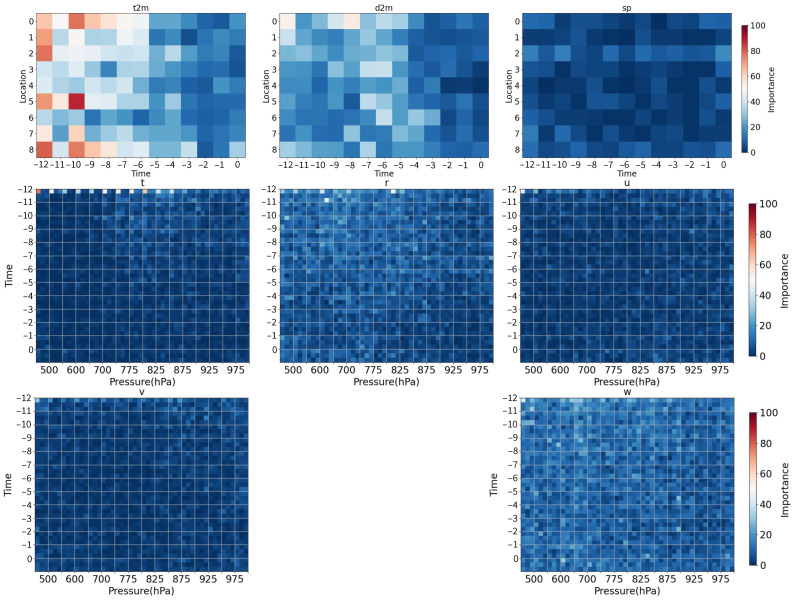
Average importance of the input features. Abbreviations correspond to those mentioned in Section 3.1.

**Figure 7 entropy-25-00138-f007:**
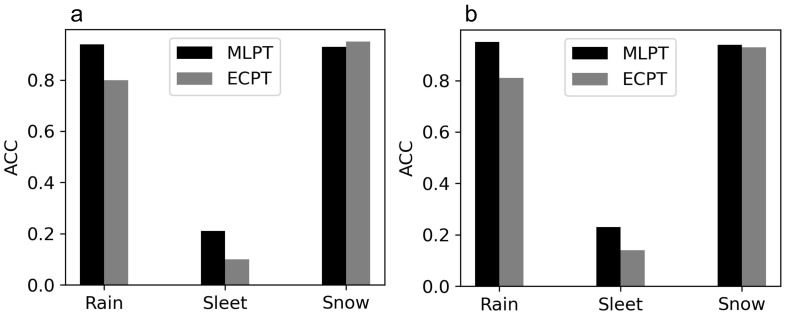
The comparison of ACC of MLPT and ECPT for three categories of precipitation types for all stations (**a**) and stations at elevations below 800 m (**b**).

**Table 1 entropy-25-00138-t001:** Contingency table for three-category of winter precipitation types. The letters a–i denote the numbers of occurrences of each precipitation type.

			Ground Truth	
		Rain	Sleet	Snow
	Rain	a	d	e
**Prediction**	Sleet	f	b	g
	Snow	h	i	c

**Table 2 entropy-25-00138-t002:** The detailed information of 14 different dataset combinations.

	Training Data	Validation Data
Set1–Set5	80% Random Sampling	20% Random Sampling
Set6	1998.01–2014.03	2014.11–2016.03
Set7	1998.01–2012.03 and 2014.11–2016.03	2012.11–2014.03
Set8	1998.01–2010.03 and 2012.11–2016.03	2010.11–2012.03
Set9	1998.01–2008.03 and 2010.11–2016.03	2008.11–2010.03
Set10	1998.01–2006.03 and 2008.11–2016.03	2006.11–2008.03
Set11	1998.01–2004.03 and 2006.11–2016.03	2004.11–2006.03
Set12	1998.01–2002.03 and 2004.11–2016.03	2002.11–2004.03
Set13	1998.01–2000.03 and 2002.11–2016.03	2000.11–2002.03
Set14	2000.11–2016.03	1998.01–2000.03

**Table 3 entropy-25-00138-t003:** The model performance of 14 sets and 3 categories of precipitation-type distributions of validation datasets (the numbers in bold type highlight the datasets with abnormal validation results).

	Validation		Test		Distribution		
	ACC	HSS	ACC	HSS	Rain	Sleet	Snow
Set1	0.91	0.81	0.81	0.64	650	449	2276
Set2	0.92	0.82	0.83	0.69	663	436	2276
Set3	0.92	0.82	0.82	0.65	593	459	2323
Set4	0.91	0.81	0.83	0.69	663	420	2292
Set5	0.91	0.82	0.81	0.65	633	436	2306
**Set6**	**0.79**	**0.58**	0.80	0.64	260	151	672
Set7	0.82	0.59	0.81	0.65	261	165	873
**Set8**	**0.86**	**0.72**	0.82	0.66	322	123	626
Set9	0.84	0.61	0.83	0.69	241	212	1057
Set10	0.83	0.68	0.82	0.67	435	209	798
**Set11**	0.88	**0.48**	0.82	0.67	**65**	139	1007
Set12	0.84	0.62	0.82	0.66	269	232	1234
Set13	0.86	0.59	0.82	0.67	155	135	861
Set14	0.83	0.64	0.83	0.68	195	100	392

**Table 4 entropy-25-00138-t004:** The model performance of 14 sets for stations below 800 m.

	Validation		Test	
	ACC	HSS	ACC	HSS
Set1	0.92	0.85	0.86	0.75
Set2	0.92	0.85	0.85	0.73
Set3	0.92	0.85	0.86	0.76
Set4	0.93	0.86	0.86	0.76
Set5	0.92	0.84	0.87	0.77
Set6	0.79	0.62	0.85	0.74
Set7	0.84	0.66	0.86	0.77
Set8	0.90	0.80	0.88	0.79
Set9	0.86	0.69	0.88	0.79
Set10	0.86	0.75	0.86	0.76
Set11	0.87	0.51	0.86	0.75
Set12	0.84	0.67	0.87	0.77
Set13	0.87	0.67	0.87	0.77
Set14	0.84	0.62	0.87	0.77

**Table 5 entropy-25-00138-t005:** Confusion matrix of precipitation-type predictions.

			Ground Truth	
		Rain	Sleet	Snow
	Rain	153/129	6/10	4/12
**Prediction**	Sleet	5/20	17/8	19/5
**(MLPT/ECPT)**	Snow	4/13	58/63	307/313

**Table 6 entropy-25-00138-t006:** Confusion matrix of precipitation-type predictions for stations at elevations below 800 m.

			Ground Truth	
		Rain	Sleet	Snow
	Rain	139/119	5/5	1/8
**Prediction**	Sleet	1/18	10/6	8/3
**(MLPT/ECPT)**	Snow	6/9	29/33	146/144

**Table 7 entropy-25-00138-t007:** Performance of precipitation-type prediction for other methods.

	ACC	HSS
LR	0.7061	0.5131
SVM	0.8048	0.6158
NB	0.6402	0.3731
MLP	0.7083	0.5038

## Data Availability

Not applicable.

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
