# Peer review of "Forecast of Winter Precipitation Type Based on Machine Learning Method"

_entropy, 2023, doi:10.3390/e25010138_

Round 1

Reviewer 1 Report

The authors used LightGBM to predict winter precipitation types at 32 locations in northern China. Each forecast predicts the precipitation type using  9714 weather variables. The authors achieved predictions which are more accurate than ECMWF forecast products.

Major comments:

1. Line 205-208: Set1-Set5 were split into 8:2, while Set6-Set14 were split into 8:1. Therefore, random sampling is not the only difference between Set1-5 and Set6-14. 

2. The objective of this study is to introduce a new method of predicting the precipitation types using LightGBM with 9714 input variables. However, the authors focus on the analysis of the validation set rather than evaluating the performance of the new approach. It is necessary to compare the new method with conventional machine learning methods such as logistic regression, SVMs, ANNs, etc. The authors also need to verify that a large number of input variables improves the predictive performance.

Minor comments:

Line 12-13: Abbreviations (ECPT and MLPT) are not adequate. 

Line 17: The section number starts from 0.

Line 76-78:  This sentence needs to be rewritten.

Line 95: 'per day' is not necessary.

Line 99 and 104: The same sentence is repeated twice.

Line 123; 'et al.' -> ', etc.'

Line 124 - 127: I cannot understand why those sentences are necessary.

Equation 2: What is G?

Line 145: Define the abbreviation, GOSS.

Equation 3: What are x_i, x_ij, and N_i^j?

Line 158: d is not max depth.

Equation 5: ACC -> PC, n = a + b + ... + i

Line 181: The meaning of 'the reanlaysis' is not clear.

Table 2: '1997.01-2000.03' -> '1998.01-2000.03'

Line 200: Is the experiment with 'h=0' necessary?

Table 3: Please explain 'Distribution' (validation + test?) and the numbers in bold type.

Reviewer 2 Report

Review of "Forecast of winter precipitation type based on machine learning method" by Zhang et al.

Synopsis: This study seeks to create a machine leaning based precipitation type (rain, sleet, snow) prediction model. In situ precipitation type records are used as "labels" and ERA-5 reanalysis data used to provide model predictors (temperature, relative humidity, wind speed, vertical velocities at surface and pressure levels). The machine leaning based prediction model is shown to perform better than corresponding ECMWF forecast products.

General comments:

The methodology appears to be sound, the results are encouraging, and the manuscript is generally well written although it would benefit from additional proofreading to remove typographical errors.

Specific comments:

Line 7: "direct" - > "suggest"?

Line 56: The area (from -> in) this method…

Line 87: "truth conventional meteorological data" needs more explanation (don't understand what this means).

Line 145: Insert (GOSS) at end of line

Equation (3): N, N^{j}_{l}, N^{j}_{r} not explicitly defined.

Line 158: "gtadient" -> "gradient"

Line 288: "do not have significantly improve" -> "do not significantly improve"

Line 320: "approxomately ->  "approximately"

Round 2

Reviewer 1 Report

Major comments:

The novelty of this study is the use of feature engineering (3.1) and LightGBM (3.2). However, it has not been shown that why we need to use thousands of input variables at multiple spatiotemporal scales.

Minor comments:

Line 115: Feature selection -> Feature preparation or Feature engineering

Line 121: temperature etc -> temperature, etc.

Line 153-158: Please italicize i and j.

Line 160: The equal sign is missing.

Round 3

Reviewer 1 Report

The authors have adequately answered most of my questions. I think the manuscript is ready for publication. Thanks.

Minor comment

Line 121: etc.. -> etc.